# FLUID REASONING REPRESENTATIONS

## ABSTRACT

Traditional large language models struggle with abstract reasoning tasks. By generating extended chains of thought, reasoning models such as OpenAI's o1 and o3 show dramatic accuracy improvements. However, the internal transformer mechanisms underlying this superior performance remain poorly understood. This work presents an early mechanistic analysis of how reasoning models process abstract structural information during extended reasoning. We analyze QwQ-32B on Mystery BlocksWorld – a semantically obfuscated benchmark that measures planning and reasoning capabilities. We find that QwQ gradually improves its internal understanding of actions and concepts through its extended rollouts, developing abstract representations that focus on structure rather than specific action names. Through steering experiments, we establish causal evidence that these adaptations improve problem solving: injecting refined representations from successful traces enhances accuracy, while symbolic representations can replace many specific Mystery BlocksWorld-obfuscated encodings with minimal performance loss. We therefore find that one of the factors driving reasoning model performance is in-context refinement of token representations – which we call Fluid Reasoning Representations. This provides early mechanistic interpretability into reasoning models.

## 1 INTRODUCTION

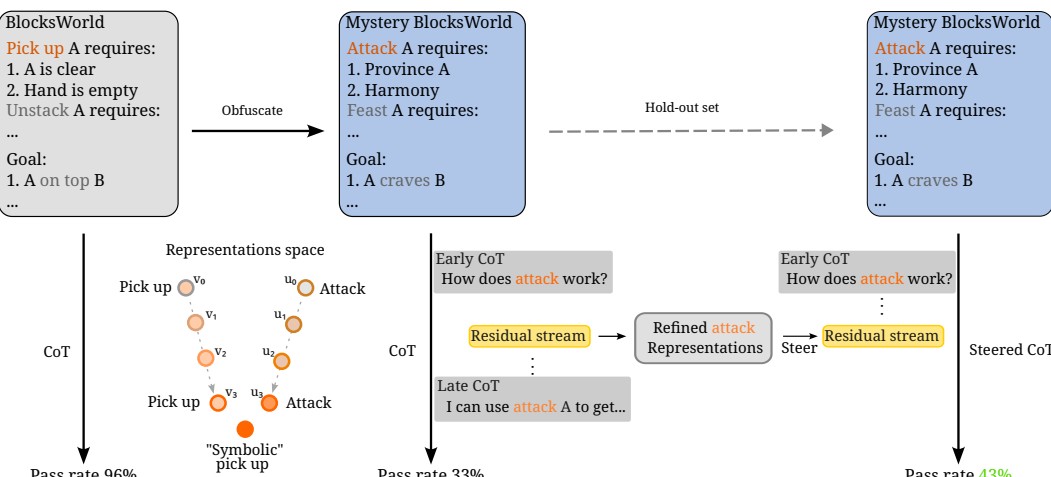

Figure 1: **Overview of our pipeline.** Left: QwQ-32B's accuracy on Standard BlocksWorld is 96%. Center: Mystery BlocksWorld obfuscates semantics (e.g., "pick up" → "attack"), reducing QwQ's accuracy to 33%. During extended reasoning traces, the model progressively refines internal representations of obfuscated actions, developing abstract symbolic encodings (vectors $v_0, \ldots, v_3$, and $u_0, \ldots, u_3$ are extracted at different Chain-of-Thought timestamps). Right: Steering experiments inject these refined representations into early reasoning stages, improving accuracy up to 43%, demonstrating that representational adaptations causally contribute to problem-solving performance.

Recent advances in large language models have produced a new class of models specifically trained to generate extended chains of reasoning before providing answers (Xu et al., 2025). These *rea-*

*soning models*, including OpenAI's o1 (OpenAI, 2024) and o3 (OpenAI, 2025), DeepSeek R1 (DeepSeek-AI, 2025), and QwQ-32B (Qwen Team, 2025), undergo extensive reinforcement learning to produce long, step-by-step reasoning traces that often span tens of thousands of tokens (Shao et al., 2024; Lambert et al., 2025). Despite their impressive capabilities, the mechanisms underlying their superior performance remain poorly understood.

Empirical evaluations reveal that these reasoning models can solve classes of problems that remain challenging for much larger traditional language models (Valmeekam et al., 2024; Shojaee et al., 2025). A striking example emerges from planning tasks where models must manipulate objects according to specific rules, such as BlocksWorld (Valmeekam et al., 2024), where the goal is to rearrange blocks to achieve target configurations. When all semantic content in these tasks is replaced with meaningless words in obfuscated versions — what we call different namings — standard LLMs achieve near-zero accuracy. For instance, transforming "pick up" into "attack" creates a new **naming** (Section 2) that strips away semantic guidance while preserving underlying logical structure. However, even moderately-sized reasoning models maintain 20-30% accuracy across different namings even when stripped of all semantic guidance (Valmeekam et al., 2024). This performance gap suggests that extended reasoning enables qualitatively new forms of structural understanding, allowing models to dynamically construct abstract problem representations during the reasoning process itself.

Despite the growing interest in understanding these capabilities, mechanistic insights into how extended reasoning traces benefit model performance remain limited. A major section of reasoning interpretability research focuses on identifying universal reasoning circuits through common token or representation-level components (Venhoff et al., 2025; Bogdan et al., 2025; Lee et al., 2025; Galichin et al., 2025). However, another possible approach is to examine the problem representations that these circuits operate on. An example of this approach is a recent work on state tracking in toy reasoning models (Zhang et al., 2025)

Prior work on in-context learning shows how models adapt internal representations when words acquire new meanings within specific contexts (Park et al., 2025). Drawing inspiration from these insights, we investigate whether similar representational adaptations occur during extended reasoning in planning tasks, and whether these adaptations causally contribute to problem-solving performance.

We focus our analysis on QwQ-32B (Qwen Team, 2025), the most capable open-source reasoning model available, and examine its internal representations while solving Mystery BlocksWorld (Valmeekam et al., 2023) puzzles. Appendices D and E also contain additional experiments on other models. Our central hypothesis is that reasoning models progressively refine their internal representations of actions and predicates during reasoning, developing context-specific semantics that enable abstract structural reasoning independent of surface-level semantics.

**Key Observations.** Our main findings about the internal mechanisms of reasoning models are:

1. **Representational Dynamics** (Section 3): We observe that QwQ-32B progressively adapts internal representations of actions and predicates during reasoning, with these adaptations converging toward consistent encodings regardless of initial action names.
2. **Causal Validation** (Section 4): Through steering experiments, we observe that these representational adaptations causally improve problem-solving performance. Injecting refined representations from successful reasoning traces into early stages of reasoning enhances accuracy on held-out puzzles, with averaged cross-naming representations achieving the strongest effects.
3. **Symbolic Abstraction** (Sections 3.2 and 4.2): We observe that adapted representations achieve symbolic abstraction, enabling cross-naming transfer. Models can operate effectively when naming-specific representations are replaced with averaged symbolic representations, suggesting convergence toward abstract structural encodings.

Our findings suggest that the superior performance of reasoning models on abstract reasoning tasks stems, at least partially, from their ability to dynamically construct problem-specific representational spaces during reasoning. This capability represents a fundamental advance in how language models process and represent abstract structural information, with implications for understanding and improving reasoning capabilities in future system. Figure 1 showcases our overall pipeline.

## 2 BACKGROUND

**BlocksWorld.** BlocksWorld is a classic planning domain from the International Planning Competitions (IPC, 1998). Each problem specifies initial and goal block arrangements, with constraints that agents can hold only one block at a time and cannot pick up blocks with others stacked above them. The domain defines four core **actions**: *pick-up*, *put-down*, *stack*, and *unstack*. The state is described using **predicates** such as *on(x,y)* (block x is on block y) or *on-table(x)* (block x is on the table). Full prompt can be found in A. We use PlanBench (Valmeekam et al., 2023) for problem generation and verification. Despite conceptual simplicity, base models fail to achieve perfect accuracy on four-block problems, while reasoning models demonstrate substantially superior performance (Valmeekam et al., 2024).

**Mystery BlocksWorld.** Mystery BlocksWorld (Valmeekam et al., 2023) replaces all predicates and actions with semantically unrelated words through alternative **namings** – systematic remappings where, for example, the action *pick-up* becomes *attack* and the predicate *on(x, y)* becomes *craves(x, y)* (prompt example in Appendix B). Each naming provides a complete semantic obfuscation that preserves the underlying logical structure while causing dramatic performance degradation. Success requires models to operate on abstract structural relationships and dynamically construct new semantic mappings from these obfuscated terms – capabilities reasoning models demonstrate significantly better than base LLMs. We generated 14 additional naming variants beyond the original, creating 15 different semantic obfuscations of the same domain structure (see Appendix H). We selected this domain because its fixed action space and strict rules provide a clear concept set for both the model and our analysis.

**Terminology.** We refer to each unique initial-goal state combination as a **puzzle** and each mapping variant as a **naming**. Our analysis focuses on 300 four-block puzzles, each mapped across all 15 mystery namings.

### 2.1 INITIAL EVALUATIONS

We conducted evaluations of various models on our BlocksWorld puzzle dataset to establish baseline performance and validate our choice of QwQ-32B for detailed analysis. They are available in Table 1.

Reasoning models consistently outperform standard LLMs on both regular and Mystery BlocksWorld tasks, though open-source reasoning models of moderate size remain limited, with DeepSeek distillation models showing particularly poor Mystery BlocksWorld performance. QwQ-32B demonstrates exceptional performance on both variants, with successful Mystery BlocksWorld solutions typically requiring 15-20k token reasoning traces – substantially longer than regular BlocksWorld problems and crucial for the semantic adaptation process investigated in this work. While Nemotron generates similarly long reasoning traces, QwQ-32B achieves superior accuracy across most mystery namings we have generated. We also provide reasoning behavior breakdown similar to Venhoff et al. (2025) in Appendix C. We mostly focus our further analysis on QwQ-32B, but also provide additional results on other reasoning models in appendices D and E to strengthen our results.

### 2.2 MYSTERY PERFORMANCE ANALYSIS

QwQ-32B's accuracy varies dramatically across mystery namings, from 0.05 to 0.47. The model performs worst on namings suggesting reversible operations ("open/close," "plant/harvest") or coherent alternative domains (legal proceedings, gardening cycles), while abstract philosophical terms, mixed sensory modalities, and semantically incoherent combinations enable better performance. This suggests that semantically connected actions and predicates make it much harder for the model to abstract away from their initial meanings. To verify this hypothesis, we generated several additional naming variants beyond the original 15, though these are not included in most experiments (see Appendix I for all results). Our steering experiments on the early layers (4.1 also suggest a connection between the semantics of the replacement words and the final performance.

Table 1: Performance comparison of models on BlocksWorld and **naming 1** Mystery BlocksWorld puzzles (all 300). "Accuracy Preserved" indicates the percentage of accuracy retained. "Tokens" represents the average length of the CoT. *Llama Nemotron computes the correct answer more often, but often answers with incorrect formatting which the evaluation suite cannot parse.*

| Model | BlocksWorld | | Mystery | | Accuracy Preserved |
|---|---|---|---|---|---|
| | Acc | Tokens | Acc | Tokens | |
| **Regular LLMs** | | | | | |
| GPT-4.1 (CoT) | 0.92 | 556 | 0.18 | 3837 | **20%** |
| Qwen2.5-32B | 0.21 | 71 | 0.00 | 1390 | **0%** |
| Qwen2.5-32B-Instruct (CoT) | 0.38 | 353 | 0.00 | 1479 | **0%** |
| Llama 3.3 70B Instruct (CoT) | 0.40 | 760 | 0.02 | 1142 | **5%** |
| **Reasoning Models** | | | | | |
| DeepSeek-R1-Distill-Qwen-32B | 0.81 | 2387 | 0.08 | 8500 | **10%** |
| DeepSeek-R1-Distill-Llama-70B | 0.66 | 2674 | 0.10 | 10636 | **15%** |
| Llama Nemotron Super 49B v1 | 0.48* | 1162 | 0.19 | 9200 | **40%** |
| QwQ-32B | 0.96 | 3633 | 0.35 | 16186 | **36%** |

As a special case, *Mystery naming 3* uses random strings, causing the model to explicitly recognize the task as BlocksWorld and directly map obfuscated terms to actions, as evidenced by substantially shorter reasoning traces ( 2k vs 15-20k tokens) and manual trace analysis. This suggests that some version of Mystery BlocksWorld was present in QwQ-32B's training data. Since we do not observe the model recognizing the BlocksWorld domain in reasoning traces for other namings, we believe it genuinely attempts to discover solutions rather than retrieving them from memory in those cases. We exclude naming 3 from representational analyses as it bypasses the semantic adaptation process under investigation.

## 2.3 REPRESENTATION COLLECTION

We follow methodology of Park et al. (2025) to collect representations of actions and predicates from reasoning traces. To collect representations of an action $a$, we first create a set of all possible token sequences that could encode this action (it may contain several tokens).

Given a timestamp (a token index) $T$, batch $b$ of reasoning traces and action $a$ we collect representations of action $a$ at the layer $L$ on a timestamp $T$ in the following manner: we select tokens on positions $[T - w, T]$ from each of the traces, where $w$ is a token window size. Then we leave only those that correspond to the token sequences associated with $a$. We also include a token right before each action (it often stores an important part of the representation). Then we take hidden states at layer $L$ for all of the token sequences. Average them across each sequence, and then average them across the batch.

For each Mystery naming $N$ we collect **in-naming** representation for each action and predicate on all layers $L$ and several timestamps. We also create centered action (or predicates) representations, by subtracting the mean of action (or predicates) representations in a given naming, following (Venhoff et al., 2025). Additionally, we create **cross-naming** representations for each action and predicate by averaging their **in-naming** centered representations across all of the namings. This operation should extract the "symbolic" part of their representations, which encodes their actual meaning in the Mystery BlocksWorld context, if it is present.

## 3 REPRESENTATIONAL STUDIES

Our main hypothesis is that reasoning models progressively refine their internal representations of actions and predicates during extended reasoning. We call such refined representations Fluid Reasoning Representations, named after fluid reasoning in humans (Ferrer et al., 2009; Wu et al., 2025). This process develops context-specific semantics that enable abstract structural reasoning independent of surface-level word meanings. We test this hypothesis by analyzing how QwQ-32B's (Qwen

Team, 2025) representations of actions and predicates evolve while solving Mystery BlocksWorld (Valmeekam et al., 2023) puzzles. Appendices D and E contain similar analysis for other reasoning models.

## 3.1 CROSS-NAMING REPRESENTATIONAL CONVERGENCE

If our hypothesis is correct, then semantically equivalent actions should converge to similar internal encodings across different mystery namings, regardless of their surface-level differences.

As a first step to investigate our hypothesis, we extract **in-naming** representations from Mystery naming 1 at timestamps 2k, 4k, 7k, and 10k tokens, then compute cosine similarities between these and centered representations from all timesteps across all other mystery namings, averaging the results. On Figure 2, we plot two lines for each Mystery 1 timestamp: one for the average similarity of an action with corresponding action from other namings and another one for average similarities of the action with different actions from the other namings. The figure shows that except for timestamp 2k cross-naming similarity increases substantially during reasoning, plateauing around 7,000 tokens — typically coinciding with the transition from experimenting with single actions to plan formulation attempts (Appendix C).

We also observe that similarities with different actions are always lower than with the corresponding ones. Relatively high ($\approx 0.2$) similarity is caused by representations of "stack" and "unstack" being closer to each other, than to "pick up" and "put down".

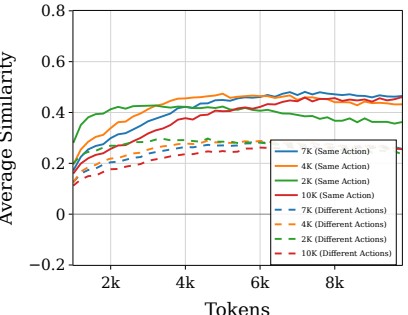

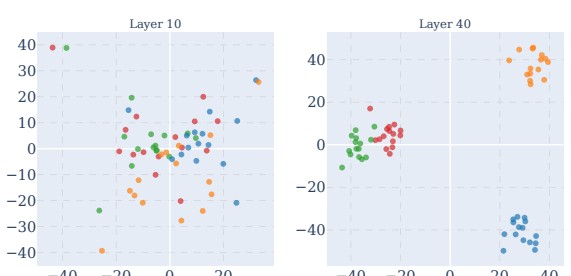

Figure 2: Average similarity of representations from other namings with naming 1 representations, extracted from different timestamps.

Figure 3: Layer-wise PCA of action representations from different mystery namings extracted at 7k tokens. More layers is Appendix D.

To visualize how representations cluster across namings, we perform PCA analysis on action representations extracted at 7k tokens from layers 10 and 40. Figure 3 demonstrates that semantically equivalent actions cluster together regardless of their surface-level naming, with clustering becoming apparent in deeper layers.

## 3.2 SIMILARITY WITH AVERAGE AND ORIGINAL BLOCKSWORLD

To better understand the nature of representational convergence, we examine similarities between naming-specific representations and average representations computed across all namings. This analysis reveals two important patterns that were obscured in the pairwise comparison.

First, when comparing centered representations with their corresponding average representations (Figure 4a), similarity increases substantially during reasoning, plateauing around 7,000 tokens. Crucially, similarities between different actions become increasingly negative as reasoning progresses. This shows the model actively differentiates between action types while developing shared encodings for equivalent actions across namings.

Second, we compare mystery naming representations at 7k tokens with clean BlocksWorld representations across all timestamps (Figure 4b). Similarity with clean BlocksWorld starts near zero and increases substantially as clean reasoning progresses. This shows the model develops simi-

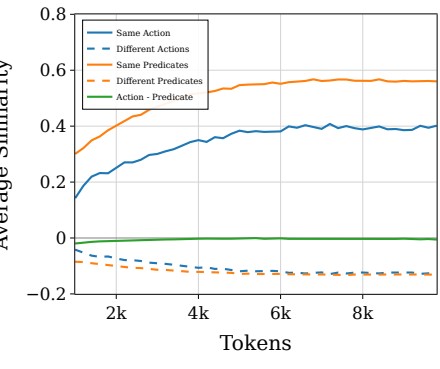
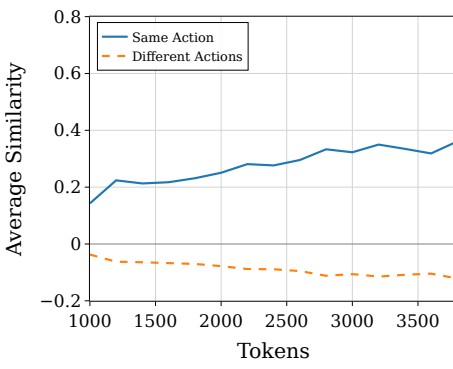

(a) Similarities with average representations    (b) Similarities with clean BlocksWorld representations

Figure 4: **Similarity with cross-naming representations between Mystery and Original BlocksWorld traces.** (a) Shows average similarities of centered action/predicate representations from all timestamps in Mystery Blocksworld traces with cross-naming representations extracted at 7k tokens. Note that similarities between different actions become increasingly negative. (b) Shows average similarities of clean BlocksWorld representations from all timestamps with cross-naming representations extracted at 7k tokens. Plot for predicates is absent, since it's much harder to identify their tokens in regular BlocksWorld traces.

lar symbolic representations even with preserved semantic content, indicating that representational adaptation is a fundamental reasoning mechanism, not just compensation for semantic obfuscation.

## 3.3 BASE MODEL COMPARISON

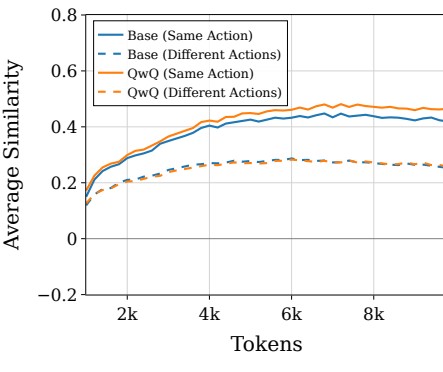
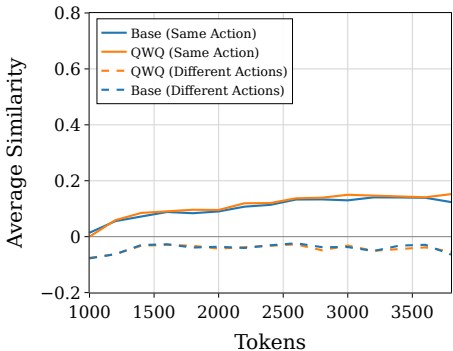

(a) Cross-naming similarity comparison    (b) Clean BlocksWorld similarity comparison

Figure 5: Average similarity of representations extracted from the 7k timestamp, plotted for both QwQ and its base model on QwQ traces. (a) Shows similarity of representations from other namings with naming 1 representations (averaged across all other namings). (b) Shows similarity of representations from original BlocksWorld traces with representations from different mystery namings (averaged across them).

We tested whether representational adaptation is specific to reasoning models by analyzing both QwQ and its base model processing identical QwQ-generated traces. Both models exhibit similar adaptation dynamics (Figure 5a), with the base model adapting slightly more slowly - likely due to processing unnatural traces. Both show comparable convergence toward shared symbolic representations (Figure 5b).

This finding, combined with prior work on in-context learning (Park et al., 2025), indicates representational adaptation is an inherent property of large language models rather than a specialized

reasoning model feature. The difference is that reasoning models naturally produce the extended context needed to use these adaptations.

# 4 CAUSAL VALIDATION

The representational analysis in Section 3 reveals that QwQ-32B dynamically adapts representations of actions and predicates beyond their original lexical meanings, with adaptations appearing independent of original word semantics.

This suggests two testable hypotheses: **(1)** representational adaptations reflect genuine improvements in understanding abstract puzzle structure, and **(2)** adapted representations achieve symbolic abstraction that transcends original tokens, enabling transfer across naming schemes. We design steering experiments to test whether learned representations contain actionable structural knowledge and can function independently of their linguistic context.

## 4.1 POSITIVE STEERING

**Experimental Setup.** Our steering procedure selects a steering layer $L$, token window $[t_{start}, t_{end})$, and steering scale $s$. We collect three types of steering vectors at layer $L$ from the 40 correctly solved puzzles: **(1)** centered **in-naming** representations $\mathbf{v}_{naming}[a]$ for all actions and predicates, **(2)** **cross-naming** representations $\mathbf{v}_{avg}[a]$ across all namings, and **(3)** random Gaussian vectors $\mathbf{v}_{rand}[a]$ scaled to match **in-naming**. We extract prefixes of $t_{end}$ tokens from a hold-out set of 100 different 4-block problem rollouts as our intervention dataset.

For each prefix $p$, we identify token indices $i$ corresponding to action or predicate $a$, obtain hidden states $\mathbf{h}$ at layer $L$, and apply the following norm-preserving intervention:

$$\mathbf{h}'[i] = s \cdot \mathbf{h}[i] + (1-s) \cdot \mathbf{v}_{type}[a], \tag{1}$$

$$\mathbf{h}[i] = \mathbf{h}'[i] \cdot \frac{\|\mathbf{h}[i]\|_2}{\|\mathbf{h}'[i]\|_2}, \tag{2}$$

where $\mathbf{v}_{type} \in \{\mathbf{v}_{naming}, \mathbf{v}_{avg}, \mathbf{v}_{rand}\}$ depending on the experiment condition. This procedure adds the refined representation while preserving activation magnitude. We measure accuracy improvement on steered puzzles compared to non-steered baseline. We selected scale $s = \frac{2}{3}$ after a sweep on layer 20 using **in-naming** representations (Appendix J.2). The steering window is [1500, 2500]. Appendix L contains improvement statistical test results.

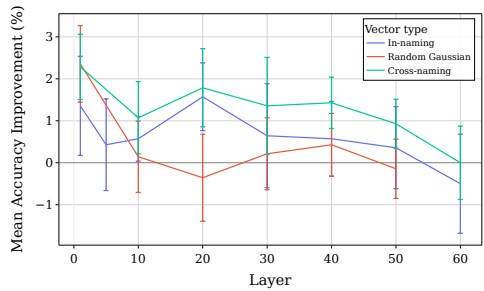
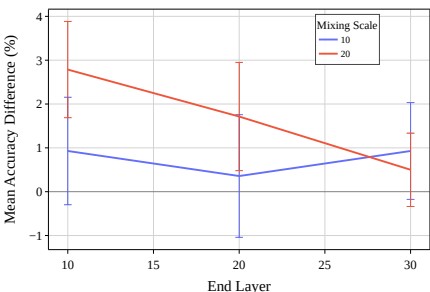

Figure 6: Accuracy improvement after positive steering averaged across mystery namings (excluding Naming 3). **Takeaways:** (i) Even random early-layer interventions ($L \leq 10$) already improve accuracy, suggesting they help remove surface-level naming associations. (ii) From $L \geq 20$ onward, steering with refined representations is most beneficial: *cross-naming > in-naming ≫* random. Error bars show s.e. across namings. See Sec. 4.1.

Figure 7: Mean accuracy difference between symbolic patching and shuffled control across scaling factors $s$. **Takeaways:** Matched *symbolic* representations outperform the shuffled baseline. This supports the *symbolic abstraction* hypothesis: the model can operate when naming-specific activations are replaced by naming-agnostic symbolic vectors. Error bars show s.e. across namings. See Section 4.2.

**Results.** Figure 6 shows accuracy improvements after positive steering, averaged across namings (excluding naming 3). Even random noise improves accuracy on early layers, suggesting that in this

case disrupting original semantic associations helps overcome interference from misleading word meanings, which is consistent with our initial evaluations (Section 2.2).

Steering with **in-naming** representations improve performance with a notable drop before layer 20, where PCA analysis reveals the onset of action representation separation (Appendix D). This suggests accuracy improvements from layer 20 onward stem from representations becoming genuinely meaningful rather than noise effects.

**Cross-naming** representations achieve the highest impact across all layers, reinforcing that these adaptations encode abstract problem structure rather than naming-specific artifacts. Together, these results support our hypothesis that learned adaptations contain meaningful structural understanding.

We also note a significant layer dependence for steering efficiency with different namings. This, along with some namings being much less responsive to steering, may be one of the reasons why average accuracy boost is relatively low (compared to gains of up to **10%** in some cases (Appendix I)).

## 4.2 Symbolic Patching

To test the **Symbolic Abstraction Hypothesis**, we conduct a patching experiment that replaces naming-specific representations with abstract "symbolic" representations and tests whether the model can operate effectively without connection to original tokens.

**Symbolic Representation Construction.** We construct symbolic representations to be minimally out-of-distribution while capturing abstract structural information. We collect centered **average** representations for each action and predicate across all namings, compute the overall mean $\mathbf{r}_{\text{mean}}$ across all actions (and separately for predicates) from all domains, and construct symbolic representations as:

$$\mathbf{r}_{\text{symbolic}}[a] = \mathbf{r}_{\text{mean}} + s \cdot \mathbf{r}_a \tag{3}$$

where $s$ is a mixing scale and $\mathbf{r}_a$ is the centered average representation of action $a$.

**Experimental Design.** Since the model maintained reasonable accuracy even when all actions were replaced with a single vector, we use a comparative approach: **(1)** Symbolic Patching replaces residual stream activations for action/predicate tokens with corresponding symbolic representations, and **(2)** Shuffled Patching uses randomly permuted symbolic representations as control.

We patch token window [2000, 4000] on all layers until the selected end layer, then measure accuracy difference $\text{Acc}_{\text{symbolic}} - \text{Acc}_{\text{shuffled}}$. Figure 7 confirms that properly matched symbolic representations consistently outperform shuffled ones across scaling factors, supporting meaningful symbolic abstraction.

## 4.3 Negative Steering

To further validate the **Structural Understanding Hypothesis**, we conduct an ablation experiment testing whether disrupting representational adaptations decreases accuracy. Since steering interventions can easily degrade performance through general disruption rather than targeted ablation, we use a comparative approach.

**Experimental Design.** We perform interventions across token window [2000, 4000] on multiple layers, subtracting centered naming representations extracted from the 4k timestamp (selected as these are near convergence while at our window's end, see Figure 2). We use shuffled representations as control, as random vectors provided insufficient baseline strength.

With optimal layer selection, negative steering shows 2.9% accuracy mean difference with control (full results in Appendix G). This reinforces that representational adaptations play a crucial role in problem-solving, as disrupting learned representations leads to measurably worse performance even when controlling for general intervention effects..

## 5 RELATED WORK

**Interpretability of Language Models' Representations.** Recent mechanistic interpretability research has converged on identifying meaningful directions in model representation spaces. Studies have demonstrated that large language models encode diverse features as linear directions in their activation spaces, including truthfulness (Li et al., 2023; Azaria & Mitchell, 2023; Marks & Tegmark, 2024; Zou et al., 2025), sentiment (Tigges et al., 2024), sycophancy (Perez et al., 2023; Panickssery et al., 2024; Sharma et al., 2024), factual knowledge (Gurnee & Tegmark, 2024), and refusal behavior (Arditi et al., 2024). Complementing these supervised approaches, sparse autoencoders have emerged as powerful tools for discovering feature directions in an unsupervised manner, revealing interpretable features at scale (Bricken et al., 2023; Huben et al., 2024; Templeton et al., 2024). These findings support the linear representation hypothesis; i.e., that neural networks encode semantic concepts as linear directions in high-dimensional activation spaces (Mikolov et al., 2013; Bolukbasi et al., 2016; Elhage et al., 2021; Nanda et al., 2023; Park et al., 2024; Olah, 2024). Beyond concept-based representations, single directions may also contain complex functional and structural information, such as generalized task definitions from in-context learning examples (Todd et al., 2024; Hendel et al., 2023), new meanings of words in in-context learning (Park et al., 2025), user-specified instructions (Stolfo et al., 2025), or even reasoning behavior itself (Zhao et al., 2025).

**Reasoning Interpretability.** A major part of reasoning interpretability research focuses on identifying universal reasoning circuits through common reasoning components. These can be key intermediate sentences or "thought anchors" (Bogdan et al., 2025), reasoning behaviors like uncertainty expression and backtracking (Gandhi et al., 2025; Venhoff et al., 2025), self-verification directions (Lee et al., 2025), reasoning-related sparse autoencoder features (Galichin et al., 2025). As an alternative representations-based approach, Zhang et al. (2025); Hou et al. (2023) study state tracking or contents of a reasoning tree in toy transformers, while Arefin et al. (2025) looks at representations during reasoning from a compression perspective. Dutta et al. (2024) investigates attention patterns and shifts of representations spaces during different reasoning behaviors representing a mix of both approaches. Finally, Ward et al. (2025) compares representations in reasoning and base models, finding that reasoning-finetuning repurposes directions already present in base model activations.

## 6 LIMITATIONS

We focus on a single reasoning model (QwQ-32B) and a single domain (BlocksWorld). While this does not cover the full diversity of reasoning tasks, BlocksWorld offers a particularly clean testbed: it has a small, well-defined set of actions and predicates, clear structural rules, and easily controlled obfuscations. This makes it possible to isolate representational adaptation in a way that would be difficult in domains with unconstrained concept spaces, such as open-ended mathematics or natural language reasoning. We expect these findings to generalize to other structured planning setups with fixed action spaces (e.g., Towers of Hanoi), though verifying this and testing whether the patterns extend to less constrained domains remains future work.

Our steering and patching intervention methods are deliberately simple, chosen to remain tractable on reasoning traces that often span 15–20k tokens. More targeted or fine-grained causal tools could sharpen the picture, but even our coarse interventions reveal measurable effects. Similarly, computational limits prevented extensive hyperparameter sweeps or decoding strategy comparisons, yet the observed representational trends were consistent across multiple obfuscations. We also note that shuffled-control experiments (Appendix K) reveal unexpected gains around later layers (notably layer 30), suggesting that some aspects of late-layer representational dynamics remain to be explained in future work.

## 7 CONCLUSION

This work analyzed how a reasoning-oriented language model (QwQ-32B) processes abstract structural information during extended reasoning. We presented three main observations. First, the model progressively refines internal representations of actions and predicates over long reasoning traces, converging toward abstract encodings that are less dependent on surface-level semantics. Second, steering experiments suggest that these representational adaptations are not merely descriptive but

can causally influence problem-solving performance: injecting refined representations tends to increase accuracy, while disrupting them tends to decrease it. Third, we observed evidence of symbolic abstraction, where representations transfer across different obfuscated namings, suggesting a degree of naming-invariant structural encoding.

Taken together, these results suggest that the superior performance of reasoning models on abstract reasoning tasks may stem in part from their ability to dynamically construct context-specific representational spaces during extended reasoning. While preliminary, our findings highlight representational refinement as a promising direction for understanding the internal mechanisms of reasoning models and contribute to a growing body of work on the interpretability of long-form reasoning traces.

## REPRODUCIBILITY STATEMENT

To ensure reproducibility of our findings, we provide implementation details throughout the paper and appendix. Model evaluation and mystery results are specified in Section 2.1 and Appendix I. Our representation extraction methodology is described in Section 3, including token window selection, averaging procedures, and timestamp choices. Steering experiment configurations, including intervention scales and layer selections, are documented in Section 4. Implementation specifics for our vLLM-based steering engine and computational considerations are provided in Appendix J.1. All 15 Mystery BlocksWorld naming variants used in our experiments are listed in Appendix H, and we commit to releasing our implementation code, experimental configurations, and processed datasets upon publication to facilitate replication and extension of our work.

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

## A BLOCKSWORLD PROMPT EXAMPLE

```
I am playing with a set of blocks where I need to arrange the blocks into
    stacks. Here are the actions I can do

Pick up a block
Unstack a block from on top of another block
Put down a block
Stack a block on top of another block

I have the following restrictions on my actions:
I can only pick up or unstack one block at a time.
I can only pick up or unstack a block if my hand is empty.
I can only pick up a block if the block is on the table and the block is
    clear.
A block is clear if the block has no other blocks on top of it and if the
    block is not picked up.
I can only unstack a block from on top of another block if the block I am
    unstacking was really on top of the other block.
I can only unstack a block from on top of another block if the block I am
    unstacking is clear.
Once I pick up or unstack a block, I am holding the block.
I can only put down a block that I am holding.
I can only stack a block on top of another block if I am holding the
    block being stacked.
I can only stack a block on top of another block if the block onto which
    I am stacking the block is clear.
Once I put down or stack a block, my hand becomes empty.
Once you stack a block on top of a second block, the second block is no
    longer clear.

Here is an example problem:

[STATEMENT]
As initial conditions I have that, Block B is clear, Block C is clear,
    the hand is empty, Block C is on top of Block A, Block A is on the
    table, Block B is on the table.
My goal is to have that Block A is on top of Block C and Block B is on
    top of Block A

My plan is as follows:

[PLAN]
unstack Block C from on top of Block A
put down Block C
pick up Block A
stack Block A on top of Block C
pick up Block B
stack Block B on top of Block A
[PLAN END]
```

## B MYSTERY PROMPT EXAMPLE

```
I am playing with a set of objects. Here are the actions I can do:
    Attack object
    Feast object from another object
    Succumb object
    Overcome object from another object

I have the following restrictions on my actions:
    To perform Attack action, the following facts need to be true:
        Province object, Planet object, Harmony.
```

```
        Once Attack action is performed the following facts will be true:
            Pain object.
        Once Attack action is performed the following facts will be false:
            Province object, Planet object, Harmony.
        To perform Succumb action, the following facts need to be true: Pain
            object.
        Once Succumb action is performed the following facts will be true:
            Province object, Planet object, Harmony.
        Once Succumb action is performed the following facts will be false:
            Pain object.
        To perform Overcome action, the following needs to be true: Province
            other object, Pain object.
        Once Overcome action is performed the following will be true: Harmony
            , Province object, Object Craves other object.
        Once Overcome action is performed the following will be false:
            Province other object, Pain object.
        To perform Feast action, the following needs to be true: Object
            Craves other object, Province object, Harmony.
        Once Feast action is performed the following will be true: Pain
            object, Province other object.
        Once Feast action is performed the following will be false:, Object
            Craves other object, Province object, Harmony.

Here is an example problem:
[STATEMENT]
As initial conditions I have that, province Block B, province Block C,
    harmony, Block C craves Block A, planet Block A, planet Block B.
My goal is to have that Block A craves Block C and Block B craves Block A
    .
My plan is as follows:
[PLAN]
feast Block C from Block A
succumb Block C
attack Block A
overcome Block A from Block C
attack Block B
overcome Block B from Block A
[PLAN END]
```

## C  BEHAVIOR ANALYSIS

Through manual investigation of DeepSeek and QwQ reasoning traces, we identified recurring be-
havioral patterns in Mystery BlocksWorld solving. Models begin with **comparative analysis**, exam-
ining initial and goal states to identify conflicting predicates. They then alternate between **recursive
search** (working backwards from goals to identify required actions) and **exploration** (experiment-
ing with actions to discover achievable states). These exploratory behaviors occupy the first half
of reasoning traces. The second phase involves **plan formulation**, where models construct action
sequences and verify validity, iteratively rebuilding when conflicts arise. The final phase consists of
**plan verification**, where models validate solutions before committing to answers.

# D  LAYER-WISE PCA

fig. 8 contains layer-wise PCA for action representations of different reasoning models. The clustering patterns become more pronounced in deeper layers, with clear separation between action types emerging around layers 20-30.

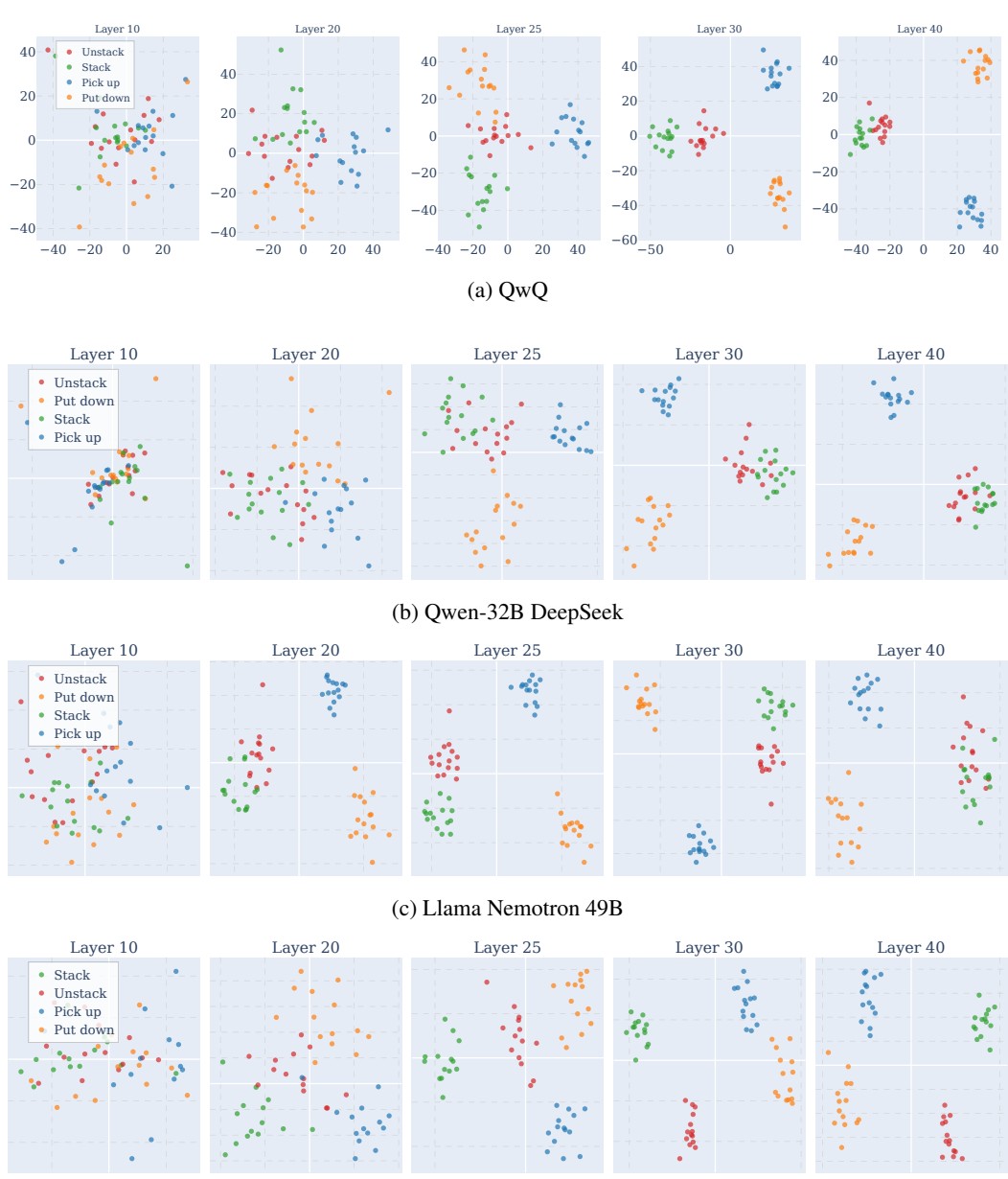

Figure 8: Layer-wise PCA of action representations from different mystery namings extracted at 7k tokens for (a) QwQ, (b) Qwen-32B DeepSeek, (c) Llama Nemotron 49B and (d) Seed-OSS-36B-Instruct

# E  CROSS-MODEL SIMILARITY ANALYSIS

To validate that representational convergence is not unique to QwQ-32B, we analyzed action and predicate representations across multiple reasoning models. Figure 9 shows how centered action

and predicate representations from different timestamps converge toward cross-naming average representations extracted at 7k tokens, analogous to the analysis in Section 3.2.

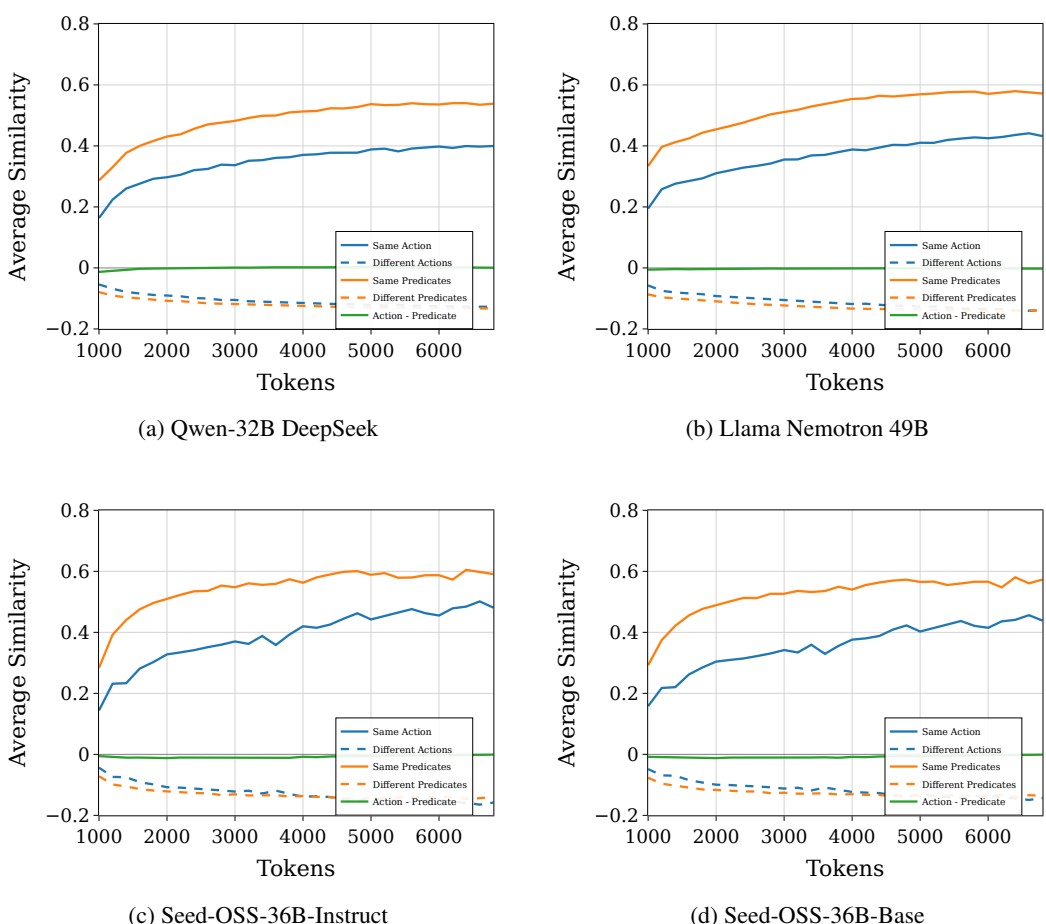

(a) Qwen-32B DeepSeek

(b) Llama Nemotron 49B

(c) Seed-OSS-36B-Instruct

(d) Seed-OSS-36B-Base

Figure 9: Similarities with cross-naming average representations across different reasoning models. Each plot shows average similarities of centered action/predicate representations from all timestamps in Mystery BlocksWorld traces with cross-naming representations extracted at 7k tokens for (a) Qwen-32B DeepSeek, (b) Llama Nemotron 49B, (c) Seed-OSS-36B-Instruct, and (d) Seed-OSS-36B-Base (ByteDance-Seed-Team, 2025). All models exhibit progressive convergence toward cross-naming average representations, with similarity increasing substantially during reasoning and plateauing around 7k tokens. Similarities between different actions become increasingly negative as reasoning progresses, demonstrating that representational adaptation is a general property of extended reasoning rather than model-specific behavior. Similarity growth for Seed-Base is slightly slower than for Seed-Instruct.

# F    HYPERPARAMETERS AND EXPERIMENTAL CONFIGURATION

This section provides a comprehensive overview of all hyperparameters and experimental configurations used throughout our analysis.

## F.1    REPRESENTATION COLLECTION

- **Token window size** ($w$): Used to identify action/predicate tokens around each timestamp. 200 is selected to contain at least 10 action mentions across the selected batch size.
- **Extraction timestamps**: 2k, 4k, 7k, 10k tokens

- **Batch size** ($b$): Number of reasoning traces per batch for representation extraction. 40 selected, since average mystery accuracy for QwQ was 0.2, giving 40 correctly solved puzzles from a 200-puzzle dataset.
- **Layers analyzed**: All layers from 0 to model depth
- **Number of puzzles for representation collection**: 40 correctly solved puzzles for steering vectors, full dataset for general analysis
- **Layer selection for representation analysis**: We selected layer 40 as a layer on which the separation of action representation converged and does not change much later. We also preformed analysis on layer 30, which did not lead to any significantly different results, so we did not include it.

## F.2 Positive Steering

- **Steering scale** ($s$): $\frac{2}{3}$ (selected after sweep, see Figure 10)
- **Steering window**: $[t_{\text{start}}, t_{\text{end}}) = [1500, 2500]$ tokens
- **Layers tested**: 1, 5, 10, 20, 30, 40, 50, 60
- **Intervention dataset**: 100 held-out 4-block problem rollouts
- **Steering vector extraction timestamp**: 7k tokens (for layer-wise analysis)

## F.3 Symbolic Patching

- **Patching window**: $[2000, 4000]$ tokens
- **End layers tested**: Multiple layers up to selected end layer
- **Scaling factors** ($s$): $\{10, 20\}$
- **Symbolic representation construction**: $\mathbf{r}_{\text{symbolic}}[a] = \mathbf{r}_{\text{mean}} + s \cdot \mathbf{r}_a$
- **Control condition**: Shuffled symbolic representations (random permutation)

## F.4 Negative Steering

- **Intervention window**: $[2000, 4000]$ tokens
- **Representation extraction timestamp**: 4k tokens
- **Start layer**: 10
- **End layers tested**: 20, 30
- **Control condition**: Shuffled centered naming representations

## F.5 Model Inference

- **Decoding strategy**: Greedy decoding
- **Maximum sequence length**: 24,576 tokens
- **Temperature**: 0 (greedy)
- **Implementation**: vLLM v0.7.3 with PyTorch forward hooks

## F.6 Dataset Configuration

- **Number of puzzles**: 300 four-block BlocksWorld puzzles
- **Number of mystery namings**: 15 (primary experiments), 20 (including additional variants)
- **Naming 3 exclusion**: Excluded from representational analyses (recognized as BlocksWorld)
- **Train/test split**: 40 correctly solved puzzles for steering vector extraction, 100 held-out puzzles for steering evaluation

# G    NEGATIVE STEERING

To further validate the Structural Understanding Hypothesis, we conduct an ablation experiment testing whether disrupting representational adaptations decreases accuracy. Since steering interventions can easily degrade performance through general disruption rather than targeted ablation, we use a comparative approach.

We perform interventions across token window [2000, 4000] on multiple layers, subtracting centered naming representations extracted from the 4k timestamp. We use shuffled representations as control, as random vectors provided insufficient baseline strength. We start steering on layer 10 and perform two runs: 1) End layer 20 gives 2.3%±0.99% difference with random. 2) End layer 30 gives 2.9%±1.06%.

# H    MYSTERY BLOCKSWORLD NAMING VARIANTS

Table 2: Action Mappings Across Mystery Namings

| Naming | pick up | put down | stack | unstack |
|---|---|---|---|---|
| Mystery 1 | attack | succumb | overcome | feast |
| Mystery 2 | illuminate | silence | distill | divest |
| Mystery 3 | tltezi | jchntg | deesdu | xavirm |
| Mystery 4 | swim | fire | deduct | respond |
| Mystery 5 | whisper | calculate | orbit | navigate |
| Mystery 6 | decode | hibernate | thunder | quench |
| Mystery 7 | explore | ripen | weave | bloom |
| Mystery 8 | harvest | ignite | carve | suspend |
| Mystery 9 | construct | demolish | reinforce | collapse |
| Mystery 10 | plant | harvest | nurture | prune |
| Mystery 11 | prosecute | acquit | testify | appeal |
| Mystery 12 | broadcast | receive | encrypt | decode |
| Mystery 13 | whisper | banish | entangle | unmask |
| Mystery 14 | question | resolve | interweave | liberate |
| Mystery 15 | summon | dismiss | fold | unravel |
| **Additional Naming Variants** | | | | |
| Mystery 16 | open | close | connect | disconnect |
| Mystery 17 | chop | serve | season | taste |
| Mystery 18 | release | grasp | separate | combine |
| Mystery 19 | transcend | sublimate | actualize | deconstruct |
| Mystery 20 | flixate | grample | chonder | sprill |

Table 3: Predicate Mappings Across Mystery Namings

| Naming | ontable | clear | handempty | holding | on |
|--------|---------|-------|-----------|---------|-----|
| Mystery 1 | planet | province | harmony | craves | pain |
| Mystery 2 | aura | essence | nexus | harmonizes | pulse |
| Mystery 3 | oxtslo | adohre | jqlyol | gszswg | ivbmyg |
| Mystery 4 | fever | marble | craving | mines | shadow |
| Mystery 5 | crystal | fountain | autumn | illuminates | legend |
| Mystery 6 | prism | hollow | zenith | echoes | emblem |
| Mystery 7 | fossil | dialect | equinox | fractures | symphony |
| Mystery 8 | nebula | labyrinth | mirage | captivates | cascade |
| Mystery 9 | eclipse | vintage | paradox | resonates | twilight |
| Mystery 10 | crystal | puzzle | vortex | whispers | cipher |
| Mystery 11 | nebula | molecule | anthem | silhouettes | voltage |
| Mystery 12 | horizon | compass | solstice | orbits | quantum |
| Mystery 13 | tethered | unburdened | hollow | shrouds | consuming |
| Mystery 14 | echoing | sovereign | potential | obscures | contemplating |
| Mystery 15 | suspended | timeless | interval | transcends | enveloping |
| **Additional Naming Variants** | | | | | |
| Mystery 16 | paired | single | balanced | matches | mirrors |
| Mystery 17 | plated | fresh | kitchen | simmering | marinated |
| Mystery 18 | floating | occupied | crowded | repels | avoids |
| Mystery 19 | phenomenal | unmediated | dialectical | instantiates | necessitates |
| Mystery 20 | morkled | thristy | plimmish | vexates | quorbles |

## I    MYSTERY PERFORMANCE ANALYSIS

Table 4: Performance across Mystery BlocksWorld naming variants with steering improvements. Columns with accuracy improvements display the maximum increase on layers 20, 30, 40 and 50.

| Naming Variant | Base Acc. | In-Naming Steering | Cross-Naming Steering | Semantic Description |
|----------------|-----------|--------------------|-----------------------|----------------------|
| Mystery 1 | 0.33 | +0.10 | +0.11 | Mixed violent/consumption metaphors |
| Mystery 2 | 0.47 | +0.05 | +0.05 | Abstract mystical/spiritual terms |
| Mystery 3 | 0.65 | — | — | Random strings |
| Mystery 4 | 0.25 | +0.03 | +0.03 | Mixed physical actions |
| Mystery 5 | 0.24 | -0.01 | +0.01 | Communication/navigation metaphors |
| Mystery 6 | 0.26 | +0.05 | +0.07 | Technical/elemental operations |
| Mystery 7 | 0.19 | +0.02 | +0.03 | Nature/growth cycle |
| Mystery 8 | 0.11 | +0.02 | +0.04 | Agriculture/crafting metaphors |
| Mystery 9 | 0.25 | +0.02 | +0.01 | Construction/destruction cycle |
| Mystery 10 | 0.05 | +0.09 | +0.05 | Coherent gardening domain |
| Mystery 11 | 0.14 | +0.00 | +0.02 | Legal proceedings domain |
| Mystery 12 | 0.16 | +0.06 | +0.02 | Communication technology |
| Mystery 13 | 0.48 | +0.06 | +0.06 | Dark mystical operations |
| Mystery 14 | 0.24 | +0.02 | +0.02 | Abstract philosophical inquiry |
| Mystery 15 | 0.34 | +0.04 | +0.04 | Mystical summoning/manipulation |
| **Additional Variants** | | | | |
| Mystery 16 | 0.05 | — | — | Reversible operations (open/close) |
| Mystery 17 | 0.27 | — | — | Coherent cooking domain |
| Mystery 18 | 0.07 | — | — | Physical manipulation verbs |
| Mystery 19 | 0.33 | — | — | Abstract philosophical concepts |
| Mystery 20 | 0.29 | — | — | Complete nonsense words |

Table 4 reveals several patterns supporting our hypothesis that semantic coherence impedes abstraction. Namings with coherent alternative domains (Mystery 10: gardening, Mystery 11: legal proceedings, Mystery 16: reversible operations) achieve the lowest base accuracies, while abstract or semantically incoherent combinations (Mystery 2, Mystery 13) enable superior performance.

The steering improvement data shows notable heterogeneity across namings. The maximum improvements reported here represent the best performance across layers 20, 30, 40, and 50, as different namings exhibit optimal responsiveness at different depths. Some namings (Mystery 5, Mystery

11) show minimal or no improvement from naming-mean steering, while others (Mystery 1, Mystery 10) demonstrate substantial gains. This suggests that certain semantic structures are more amenable to representational refinement than others.

## J    IMPLEMENTATION DETAILS

### J.1    STEERING ENGINE

We implement steering using PyTorch forward hooks on top of vLLM v0.7.3 v0 Kwon et al. (2023), which provides substantial performance improvements, reducing experiment runtimes from several hours to tens of minutes compared to alternatives like TransformerLens (Nanda & Bloom, 2022) or NNSight Fiotto-Kaufman et al. (2024). However, this approach has tradeoffs: we must be mindful of cache recomputations, since they recompute representations without steering interventions, and vLLM's optimizations introduce some numerical instability during extended reasoning traces. To address this instability, we run experiments across multiple naming variants. All experiments use greedy decoding with maximum sequence length of 24,576 tokens.

### J.2    HYPERPARAMTERS

We perform positive steering as described in Section 4.1 on layer 20 using **naming-mean** representations to determine the optimal steering scale. Figure 10 shows that scales $\frac{2}{3}$ and $\frac{4}{5}$ have similar effects. While improvement from $\frac{4}{5}$ is slightly higher, we chose $\frac{2}{3}$ since it has a more stable effect on all namings.

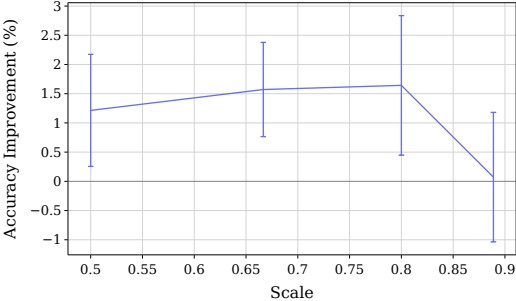

Figure 10: Positive steering results for layer 20 using different scale parameters $s$.

## K    SHUFFLED IN-NAMING STEERING (EXPLORATORY)

**Setup.**    In addition to positive steering (Section 4.1), we tested a *shuffled in-naming* control. For each naming, we applied a single consistent permutation of centered in-naming vectors across actions/predicates (e.g., pick up→stack, stack→put down, etc.). This preserves per-naming distributional statistics while breaking the action–representation alignment. Interventions used the same window $[1500, 2500]$, scale $s = \frac{2}{3}$, and norm-preserving update rule as before, run on a reduced subset of layers and namings.

**Findings.**    Figure 11 shows a three-phase trend: (i) early layers ($\leq 5$) improve relative to baseline, consistent with disruption of surface semantics; (ii) middle layers ($\sim 5 - 20$) degrade performance, likely breaking emerging abstractions; (iii) later layers ($\geq 30$) again show improvements, comparable to unshuffled *in-naming*.

**Interpretation.**    Early/mid results align with our story: disruption helps initially, but permutations harm once action-specific representations form. Late-layer gains suggest action vectors contain shared structural components, so even mismatched but in-manifold vectors can occasionally assist.

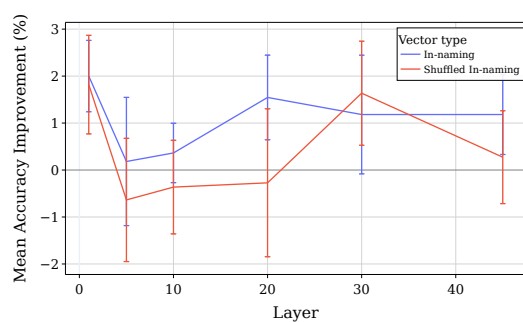

Figure 11: **Accuracy change under shuffled in-naming steering.** See Appendix K.

**Caveat.** These runs covered fewer namings/layers and late-layer effects were heterogeneous. We report them here for transparency; a fuller sweep is left to future work.

## L    STATISTICAL ANALYSIS OF STEERING EFFECTS

We conducted statistical tests to validate that steering with refined representations improves accuracy over baseline. Our analysis uses one-sample t-tests treating each mystery naming as an independent observation.

### L.1    TEST METHODOLOGY

**Data Structure.**    Our experiments evaluate steering across 14 mystery namings (excluding naming 3) with approximately 100 puzzles per naming (indices 200-300). Each mystery naming provides accuracy measurements under multiple conditions: baseline (no steering), in-naming steering, cross-naming steering, and random Gaussian steering at various layers.

**Statistical Test.**    We employ one-sample t-tests to assess whether mean accuracy improvements across mystery namings significantly exceed zero. Each mystery naming serves as an independent observation, with the null hypothesis $H_0 : \mu_{\text{improvement}} = 0$. We use one-tailed tests since we hypothesize positive improvements. The test statistic is:

$$t = \frac{\bar{\Delta}}{\text{SE}(\Delta)} = \frac{\bar{\Delta}}{s_\Delta/\sqrt{n}} \tag{4}$$

where $\bar{\Delta}$ is the mean improvement across $n = 14$ mystery namings, $s_\Delta$ is the sample standard deviation, and $\text{SE}(\Delta)$ is the standard error.

### L.2    RESULTS

Table 5 shows that several steering conditions produce statistically significant improvements over baseline. Layer 20 in-naming ($p = 0.042$), layer 20 cross-naming ($p = 0.044$), and layer 40 cross-naming ($p = 0.021$) all achieve significance at $\alpha = 0.05$, with mean improvements ranging from 1.4% to 1.8%.

Random Gaussian steering at layer 20 shows negative mean improvement ($-0.36\%$, $p = 0.627$), confirming that improvements from structured representations are not artifacts of random perturbations. Cross-naming representations show numerically higher improvements than in-naming representations at all tested layers, with the strongest and most consistent effects observed at layer 40 where cross-naming achieves 1.43% improvement ($p = 0.021$) compared to 0.57% for in-naming ($p = 0.274$).

### L.3    DISCUSSION

Our statistical analysis provides evidence that steering with refined representations improves accuracy over baseline, with the strongest effects observed at layers 20 and 40 for cross-naming steering.

Table 5: Accuracy improvements from steering interventions. Mean improvements and standard errors (SE) are computed across 14 mystery namings. Significance levels: * $p < 0.05$, ** $p < 0.01$, *** $p < 0.001$, ns = not significant.

| Condition | Mean $\Delta$ | SE | t-statistic | p-value | Sig. |
|---|---|---|---|---|---|
| Layer 20 In-naming | 1.57% | 0.84% | 1.878 | 0.042 | * |
| Layer 20 Cross-naming | 1.79% | 0.97% | 1.846 | 0.044 | * |
| Layer 20 Random Gaussian | $-0.36\%$ | 1.08% | $-0.332$ | 0.627 | ns |
| Layer 30 In-naming | 0.64% | 1.29% | 0.500 | 0.313 | ns |
| Layer 30 Cross-naming | 1.36% | 1.20% | 1.133 | 0.139 | ns |
| Layer 40 In-naming | 0.57% | 0.92% | 0.618 | 0.274 | ns |
| Layer 40 Cross-naming | 1.43% | 0.64% | 2.249 | 0.021 | * |
| Layer 50 In-naming | 0.36% | 1.01% | 0.352 | 0.365 | ns |
| Layer 50 Cross-naming | 0.93% | 0.61% | 1.531 | 0.075 | ns |

The significant improvements at multiple layers, combined with the lack of improvement from random Gaussian controls, support our hypothesis that representational adaptations during reasoning contain meaningful structural information that causally contributes to problem-solving performance.

The moderate effect sizes (1.4-1.8% for significant conditions) and variability across mystery namings reflect the challenging nature of using steering with long reasoning rollouts.

# M  USE OF LARGE LANGUAGE MODELS (LLMS)

In accordance with ICLR 2026 policy requirements, we disclose the following use of large language models in the preparation of this paper:

1. **Writing assistance**: LLMs were used to improve clarity, grammar, and overall readability of the manuscript text.

2. **Terminology generation and classification**: Claude (Anthropic) was used to generate alternative Mystery namings and assess semantic relationships between generated terms.

3. **Code generation**: LLMs were used to generate plotting code and other boilerplate code for data visualization and routine programming tasks.

