# OpenReview forum: "Fluid Reasoning Representations"
_ICLR.cc/2026/Conference — Submitted to ICLR 2026_

### Official Review · Reviewer_cJuV · 2025-10-29

**Soundness:** 2
**Presentation:** 2
**Contribution:** 3
**Rating:** 4
**Confidence:** 3

**Summary:**

This paper investigates how reasoning models develop internal representations during long chain-of-thought (CoT) reasoning. Specifically, the authors analyse the QwQ-32B model on an obfuscated planning task (Mystery BlocksWorld) to test whether reasoning generalises beyond surface-level token identities. The authors hypothesise that *reasoning models progressively refine their internal representations of problem entities during reasoning, developing context-specific semantics that enable abstract structural reasoning independent of surface-level semantics*.


The study comprises three main analyses:
1. *Representational Dynamics:* The authors track how hidden representations of actions and predicates evolve over reasoning timestamps and across multiple “naming” schemes, showing that representations of the same underlying concepts converge across different surface names.


2. *Causal Steering:* To test whether these learned representations are behaviorally meaningful rather than merely correlational, the authors perform activation steering, i.e., directly injecting or perturbing hidden-state vectors during reasoning, and observe that positive steering improves accuracy while negative steering degrades it.


3. *Symbolic Patching:* To probe abstraction, the authors replace naming-specific embeddings with averaged, naming-invariant “symbolic” representations. Performance remains stable mainly, suggesting the model’s reasoning operates in an abstract representational space.

**Strengths:**

- The paper addresses an important mechanistic question: how reasoning models internally represent abstract structure during extended reasoning.

- The experimental setup is well-motivated, and the task (Mystery BlocksWorld) provides a clean testbed for analyzing abstraction.

- The authors conduct multiple complementary analyses, including steering and patching interventions.

**Weaknesses:**

- The paper’s central hypothesis that "**reasoning models** dynamically refine internal representations of problem entities ..." requires validation across more than a single model. Demonstrating the same phenomena in at least one additional reasoning model (and ideally contrasting with a non-reasoning or base variant, such as Qwen-32B) would significantly strengthen the claim.


- The robustness and reproducibility of the results are not yet clear. Some experimental details (e.g., the choice of layers, token windows, or the 40 “solved” reference puzzles) appear somewhat arbitrary or underspecified. Including these in a reproducibility table or appendix would greatly help future replication efforts.

- The writing occasionally over-generalises the findings, implying broader conclusions than the presented evidence supports. The authors should temper claims about “reasoning models” in general and clarify that observations are currently limited to QwQ-32B and the specific task.

- Some results (e.g., steering improvements) are small and would benefit from clearer statistical reporting and effect sizes.

**Questions:**

Please check the weakness section

---

> ### Author Response · Authors · 2025-12-03
>
> We appreciate the reviewer's recognition that our work addresses an important mechanistic question with a well-motivated experimental setup and multiple complementary analyses. Below, we respond to each concern and summarize the revisions we have made.
>
> **W1:** We have now added representational analyses for the other reasoning models included in our initial evaluations (**Appendices D-E**), including DeepSeek-R1 distillations and Llama Nemotron. All of these models exhibit similar convergence patterns.
> The set of models suitable for our analysis is inherently limited: models must be (1) large enough to solve Mystery BlocksWorld reliably (models under 30B parameters achieve below 5% accuracy), (2) open-source to permit representation extraction, and (3) tractable for processing 15–20k-token reasoning traces. One additional model that satisfies these constraints is Seed OSS (49B), which we have also added to the analysis and which exhibits similar representational convergence. We believe these additions meaningfully strengthen the generality of our findings.
>
> **W2:** Thank you for this helpful suggestion. We have added a dedicated section (**Appendix F**) with hyperparameter details, including:
> * Layer selection rationale
> * Token window choices and their justification
> * Details on the selection of the 40 solved reference puzzles
> We hope this additional documentation facilitates reproducibility and provides clearer insight into our experimental design choices.
>
> **W3:** We have revised our claims throughout the paper to be more precise about the scope of our findings. We have also added representation analysis results on additional reasoning models to provide stronger support for the generality of our conclusions.
> **W4:** We have added a dedicated appendix section (Appendix L) with comprehensive statistical analysis of steering effects. We now report:
> * One-sample t-tests validating that steering improves accuracy over baseline (layer 20 cross-naming: p = 0.044; layer 40 cross-naming: p = 0.021)
> * Standard errors and degrees of freedom for all reported improvements
> * Effect sizes showing steering improvements of 1.4–1.8% for significant conditions
> * Statistical validation that random Gaussian controls show no improvement (p = 0.627), confirming that effects are not artifacts of perturbation
> The analysis treats each mystery naming as an independent observation (n = 14), providing appropriate statistical rigor for our experimental design. We acknowledge that the moderate effect sizes reflect both the challenging nature of Mystery BlocksWorld and the heterogeneity across semantic obfuscations. Nevertheless, these results demonstrate statistically significant causal effects of representational refinement on problem-solving performance.
>
> We hope these revisions address the reviewer's concerns and strengthen the paper's contribution. We are grateful for the constructive feedback

---

### Official Review · Reviewer_tJg2 · 2025-10-31

**Soundness:** 3
**Presentation:** 3
**Contribution:** 2
**Rating:** 4
**Confidence:** 3

**Summary:**

This paper analyzes how QwQ-32B's internal representations evolve when solving Mystery BlocksWorld, a semantically obfuscated planning benchmark. The authors extract representations of actions and predicates at different points during reasoning traces and show that representations converge toward similar encodings across different naming schemes. They conduct steering experiments where refined representations from successful traces are injected into new problems, and symbolic patching experiments where naming-specific representations are replaced with averaged vectors.

The paper does not contain any mathematical properties or theorems. The experimental setup and research methodology appears sound.

Overall, the paper adapts the framework in Park et al. for planning problems. It showed that the phenomenon discovered in Park et al. can also be observed in planning problems, in particular, in the Mystery Blocks World instances.

**Strengths:**

1. Clear demonstration of representational convergence: The paper effectively shows that representations become increasingly similar across namings as reasoning progresses, with divergent representations at early timestamps converging around 7k tokens. This temporal progression is well-visualized and makes the adaptation process tangible, providing genuine insight into how models refine their understanding during extended generation.

2. Causal Validation via Steering: The paper successfully uses steering experiments to prove its claims. By injecting the refined representations from successful traces into new problems, the authors show these representations causally improve problem-solving accuracy. The fact that the averaged "cross-naming" representations (which are purely abstract) had the strongest positive effect supports the paper's central hypothesis.

3. Sophisticated cross-naming methodology: Creating 15 diverse naming variants and averaging representations across them to extract symbolic encodings is a thoughtful approach to isolating abstract structural meaning from surface-level lexical information.

4. Insightful base model comparison (Section 3.3): The finding that base models exhibit similar representational adaptation when processing the same traces is valuable. This helps clarify that reasoning models leverage a fundamental capability of transformers through extended generation.

**Weaknesses:**

1. Insufficient novelty over Park et al. (2025): That models adapt representations during in-context learning is documented in prior work (Park et al., 2025, cited by authors). This paper essentially applies their framework to Blocks World rather than discovering reasoning-specific mechanisms.

2. Single model analysis despite multi-model data: Table 1 shows results for DeepSeek-R1, Llama Nemotron, and QwQ, yet only QwQ is analyzed mechanistically. This makes it unclear if the findings are general to all models or specific to QwQ. Since the authors already ran performance benchmarks, why not apply the same representational analysis to the other models to test for generalization?

3. Single domain with no generalization evidence: BlocksWorld has only a small number of concepts with deterministic rules. Zero evidence that findings extend to other reasoning domains. Testing even one additional domain is essential to demonstrate this isn't domain-specific.

4. Missing explanation of refinement mechanism: The paper documents that representations converge toward symbolic encodings but does not explain the internal mechanism by which this refinement occurs. What computations cause early representations to transform into refined ones? Which attention heads read/write these representations? What information do different layers add during refinement? The paper observes the phenomenon (representations change) without explaining the process (how/why the model performs this transformation internally).

**Questions:**

Please see the weaknesses session.

---

> ### Author Response · Authors · 2025-12-03
>
> We thank the reviewer for their questions and the opportunity to clarify these aspects of our work.
>
> **W1.** Our setting differs from Park et al. in two important ways:
>
>   * Their work uses toy in-context learning prompts with single-token concepts and no model-generated reasoning. In contrast, we study naturally generated 15-20k-token reasoning rollouts, where concepts may appear under multiple tokenizations.
>   * Their causal analysis is minimal (a single appendix experiment). We perform multiple causal interventions — positive steering, negative steering, and symbolic patching — that demonstrate effects even 10k+ tokens into reasoning.
> Accordingly, our contribution is not simply reapplying their method, but rather extending the study of representational adaptation into the extended reasoning regime.
>
> **W2.** We have now added representational analyses for the other reasoning models included in our initial evaluations **(Appendices D-E**).
> The set of models suitable for our analysis is inherently limited. Models must be (1) large enough to solve Mystery BlocksWorld reliably (models under 30B parameters achieve below 5% accuracy), (2) open-source to permit representation extraction, and (3) tractable for processing 15–20k-token reasoning traces. One model that satisfies these constraints is the Seed OSS model, which we have now included.
> We were unfortunately unable to run comprehensive steering experiments on additional models due to compute constraints, but we believe the expanded representational analyses help address this concern.
>
> **W3.** We agree that additional domains would strengthen generalization claims. However, BlocksWorld's fixed symbolic structure (4 actions, 5 predicates) is precisely what makes it well-suited for this analysis -- we can track specific concept representations over time in a controlled manner. More open-ended domains such as mathematics lack fixed symbolic vocabularies, which would require fundamentally different extraction methodologies where "evolving representations" cannot be tied to consistent tokens.
> Importantly, our 15 independent naming variations effectively test robustness: each naming provides a different semantic landscape, with accuracy ranging from 5% to 47%, yet convergence patterns remain consistent across all variations. This within-domain variation provides substantial evidence for the robustness of our findings to surface-level semantic differences.
> We position our work as studying representation adaptation in domains with fixed symbolic structure (planning, formal reasoning), which we state clearly in the Limitations section. Extension to less constrained domains remains valuable future work.
>
> **W4.** These are excellent questions that get at the heart of how reasoning models work internally. We can offers some initial insights:
> Our PCA analysis (**Figure 8, Appendix D**) reveals that action separation emerges only after layer 25 (out of 64), suggesting that roughly the first 40% of layers perform disambiguation and semantic abstraction, while deeper layers operate on the resulting representations. Interestingly, random steering improves accuracy in layers below 10, which suggests that early layers primarily extract original word semantics from tokens and that perturbations can disrupt this process.
> That said, deeper questions — such as "Which attention heads read and write these representations?" and "What specific computations cause representations to transform?" — merit dedicated investigation. Our analysis is deliberately macroscopic, averaging across tokens and rollouts to identify stable patterns amid highly variable individual token representations. Circuit-level mechanistic analysis would require fundamentally different methodology and substantial additional effort; answering these questions comprehensively could constitute an entire paper in itself. We view this as a promising direction for future work.
>
> We hope these responses address the reviewer's concerns, and we are grateful for the feedback that has helped us strengthen the paper.

---

### Official Review · Reviewer_pnSR · 2025-10-31

**Soundness:** 2
**Presentation:** 1
**Contribution:** 3
**Rating:** 2
**Confidence:** 4

**Summary:**

This paper presents an early mechanistic analysis of how reasoning models process abstract structural information during extended reasoning, which analyzes QwQ-32B on Mystery BlocksWorld. This paper finds that QwQ gradually improves its internal understanding of actions and concepts through its extended rollouts, developing abstract representations that focus on structure rather than specific action names. Through steering experiments, it establishes causal evidence that these adaptations improve problem solving.

**Strengths:**

1. This paper provides insights for abstract reasoning area.
2. The method is somewhat novel.
3. The discovered theory can be applied for LLMs enhancement.

**Weaknesses:**

1. The presentation of this paper should be significantly improved.
2. The experiments are limited, the conclusions are not universally applicable.

**Questions:**

This method about this paper is novel, while the experiments and presetations shoule be significantly improved before acceptance.

1. Figure 1 is placed after the abstract, while there is no details about Figure 1 in the introduction.
2. There is neither a formal definition of the task nor exmples of the task.
3. What does in-naming and cross-naming mean? What does high and low values of them represent?
4. There should have a formal definitation of Mystery BlocksWorld.
5. I know action and predicate in language, are they the same in your paper?
6. The analysis are about actions and predicates, why the hypothese is about entities (such as lines 203-204)?
7. "we first create a set of all possible token sequences that could encode this action". How to understand "token sequences" encode "this action"? There are many such difficult-to-understand sentences in the article.
8. Conducting expeirments on other reasoning LLMs is helpful to make your conclusions universal.
9. More datasets should be considered to further enhance the persuasiveness.
10. A whole workflow is needed to better demonstrate your method, only the text can make confusion.

---

> ### Author Response · Authors · 2025-12-03
>
> We thank the reviewer for their feedback. Below, we address each point and summarize the revisions we have made to improve clarity and accessibility.
>
> Q1, Q10: **Figure 1** is intended as a workflow overview of our pipeline (evaluation → representation extraction → steering). We have now stated this explicitly in the Introduction to help orient readers.
>
> Q2, Q4, Q5: A complete domain definition and task examples for both regular and Mystery BlocksWorld are now included in **Appendices A-B**, with references added in **Section 2**. We have also revised Section 2 to provide a clearer and more formal description of BlocksWorld and Mystery BlocksWorld, including the roles of actions, predicates, and naming conventions.
>
> Q3: "In-naming" and "cross-naming" refer to the two types of representations we construct. Their definitions and construction process appear in Section 2.3, and Figure 6 shows average accuracy improvements when steering with each representation type.
>
> Q6: The phrase "core problem entities" was intended to refer to structural components of the domain (specifically, actions and predicates). We have revised the wording to avoid ambiguity and now use "actions and predicates" consistently throughout.
>
> Q7: The "set of possible token sequences that encode action a" refers to all tokenizations of an action that may appear in a rollout. For example, the action "attack" might appear as "attack," "Attack," or as multiple tokens such as ["att", "ack"].
>
> Q8: We have added representational analyses for the other reasoning models included in the study (**Appendices D-E**). The set of models for which representation extraction is feasible is limited: a model must be large enough to solve Mystery BlocksWorld reliably (models under 30B parameters often achieve below 5% accuracy) while remaining small enough for us to extract representations from 15–20k-token reasoning traces without specialized infrastructure such as vLLM. The Seed OSS model fits these constraints and has been included. Full steering experiments require substantially more compute, which prevented us from running them on additional models at this time.
>
> Q9: We agree that broader datasets could strengthen our claims. However, more open-ended domains (e.g., mathematical reasoning) require substantially more complex representation extraction, as evolving representations are not tied to a fixed set of action or predicate tokens. Given the additional model results and the 15 Mystery naming variations we evaluate, we believe BlocksWorld provides a suitable and well-controlled setting for studying representation adaptation in domains with fixed symbolic structure.
>
> We hope these clarifications and revisions address the reviewer's concerns and improve the presentation and accessibility of the paper. We are grateful for the opportunity to strengthen our work based on this feedback.

---

### Meta-Review · Area_Chair_eKJN · 2026-01-07

**Summary:**

The paper studies an important question in mechanistic interpretability of long-form reasoning: whether and how models' internal representations of symbolic structure become less tied to surface tokens during extended CoT (using Mystery BlocksWorld as a controlled testbed).

Reviewers generally agree the core empirical phenomenon, i.e., cross-naming representational convergence over time, appears real and is complemented with multiple analyses.

The rebuttal substantially improves clarity and reproducibility concerns with formal task definition, examples, implementation details and extended results. However, even after these improvements, the concern on the narrow scope and limited external validity (one main domain; causal interventions largely on a single model) is still not fully resolved.

**Reviewer Concerns:**

I did not see active discussions beyond the authors' initial response. After reading through the rebuttal, the following concerns were addressed well in the revision:
* The additional results suggest convergence patterns also appear in additional open models, which helps with validating the convergence phenomenon.
* The revision adds statistical tests and effect sizes, even though the effect sizes remain modest

Outstanding concerns:
* The paper remains limited to a single structured domain with four actions and five predicates; the rebuttal’s justification is reasonable but it does not substitute for evidence that the phenomenon matters in broader reasoning domains.
* Average steering gains are small (and sometimes borderline in significance)
* Limited novelty relative to closely related prior work on representational adaptation (Park et al.) and steering-based reasoning interpretability

**Reviewer Scores:**

The revisions directly address most of Reviewer pnSR's concrete presentation/clarity questions. Remaining concerns about limited experiments likely won't change the overall rating.

Reviewer tJg2 and Reviewer cJuV are likely to maintain or slightly tune up their ratings. The weaknesses noted by Reviewer tJg2 on limited novelty and empirical evidence + missing refinement mechanism are only partially addressed. The remaining gap in causal generalization and modest effect sizes could keep the score near the borderline.

---

### Decision · Program_Chairs · 2026-01-26

Reject